# LPO: Towards Accurate GUI Agent Interaction via Location Preference Optimization

## Abstract

The advent of autonomous agents is transforming interactions with Graphical User Interfaces (GUIs) by employing natural language as a powerful intermediary. Despite the predominance of Supervised Fine-Tuning (SFT) methods in current GUI agents for achieving spatial localization, these methods face substantial challenges due to their limited capacity to accurately perceive positional data. Existing strategies, such as reinforcement learning, often fail to assess positional accuracy effectively, thereby restricting their utility. In response, we introduce **L**ocation **P**reference **O**ptimization (**LPO**), a novel approach that leverages locational data to optimize interaction preferences. **LPO** uses information entropy to predict interaction positions by focusing on zones rich in information. Besides, it further introduces a dynamic location reward function based on physical distance, reflecting the varying importance of interaction positions. Supported by Group Relative Preference Optimization (GRPO), **LPO** facilitates an extensive exploration of GUI environments and significantly enhances interaction precision. Comprehensive experiments demonstrate **LPO**'s superior performance, achieving SOTA results across both offline benchmarks and real-world online evaluations. Our code will be made publicly available soon.

## 1 Introduction

*"The measure of intelligence is the ability to change." — Albert Einstein*

The advent of autonomous agents has profoundly altered strategies for Graphical User Interface (GUI) interactions Zhang et al. (2024a); Lieberman (1997); Wang et al. (2024). By utilizing natural language as an intermediary Hong et al. (2023), these agents minimize labor and time costs associated with manual GUI operations, thus leading to their growing prevalence in recent times Zhang et al. (2024a).

Most GUI agents rely heavily on Supervised Fine-Tuning (SFT) during the training process Hong et al. (2023); Deng et al. (2023a); Cheng et al. (2024); He et al. (2024). However, SFT often encounters significant challenges in spatial localization due to its limited capability to perceive and interpret positional data Qin et al. (2025). This shortcoming impairs precise interactions within the GUI, highlighting the fundamental challenge of improving the accuracy of such interactions.

Despite some strategies Qin et al. (2025); Xia & Luo (2025); Lu et al. (2025); Zhang et al. (2023); Liu et al. (2025) attempting to utilize Reinforcement Learning (RL) to enhance the accuracy of UI action decisions, these methods often lack a mechanism for accurately assessing interactions' positional accuracy. As a result, their ability to improve interaction accuracy is limited (as illustrated in Figure 1 (a) & (b) & (c)). Additionally, some methods like UI-TARS Qin et al. (2025) rely heavily on manually constructing positive and negative actions for direct preference optimization, thereby becoming highly dependent on data construction. Consequently, these methods fail to fully resolve the issue of precise spatial localization during GUI interactions.

To align precise GUI interaction, we introduce **L**ocation **P**reference **O**ptimization (**LPO**), an innovative approach that leverages locational data for optimizing accurate interaction preferences. Specifically, drawing inspiration from the tendency of users to interact more frequently in zones with higher information density, we divide the interface into distinct windows and employ their information entropy to build a reward for preliminarily forecasting interaction positions (see Section 4.1). Subse-

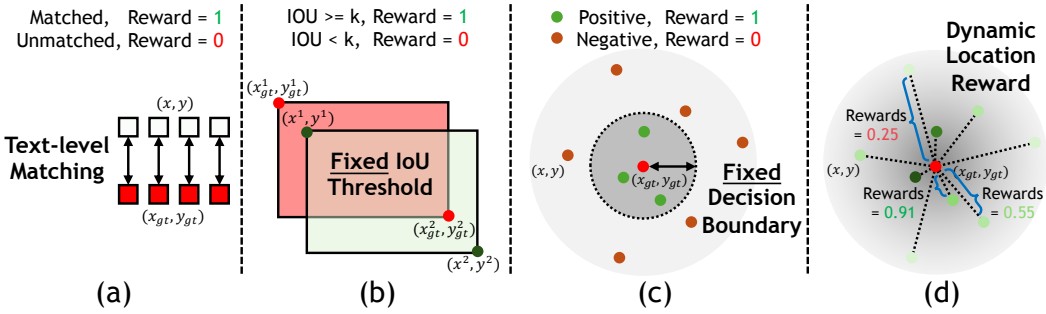

Figure 1: Motivation of dynamic location reward. **(a)** UITARS Qin et al. (2025) uses direct text-level matching; **(b)** UI-R1 Lu et al. (2025), InfiGUI-R1 Liu et al. (2025) and RUIG Zhang et al. (2023) employ bounding boxes for interaction preferences; **(c)** GUI-R1 Xia & Luo (2025) relies on fixed positional boundaries. **(d)** Our *dynamic location reward* offers a more precise positional representation, addressing the limitations of previous methods.

quently, to offer a more nuanced representation of the varying significance of interaction positions, we incorporate physical distance to develop a dynamic location reward function (see Section 4.2 and Figure 1 (d)). Finally, by integrating these rewards, we implement **LPO**, inspired by Group Relative Preference Optimization (GRPO) Shao et al. (2024). This methodology enables a more comprehensive exploration of expansive GUI environments, guiding the agent to optimize preferences that correspond to precise interaction capabilities (see Section 4.3).

Our experimental results comprehensively demonstrate that **LPO** significantly enhances the interaction capabilities of GUI agents, achieving state-of-the-art (SOTA) performance compared to other preference optimization strategies. This improvement is evident in offline benchmarks, both in GUI Interaction (Multimodal Mind2Web Deng et al. (2023b)) and Grounding (VisualWebBench Liu et al. (2024) and Screenspot V2 Wu et al. (2024)). Furthermore, our approach also exhibits superior performance in real-world scenarios during online evaluations (WebVoyager He et al. (2024)).

Our contributions can be summarized as follows:

- We design a window-based reward for predicting interaction positions, utilizing information entropy to facilitate preliminary forecasting of these locations within the GUI.
- We introduce a dynamic location reward that integrates physical distance, offering a precise representation of the varying importance associated with different interaction positions.
- Extensive experiments demonstrate that **LPO** achieves SOTA performance in GUI interaction and grounding, outperforming other baselines in both offline benchmarks and online GUI environments.

## 2 RELATED WORK

**GUI Agent Interaction**  The development of Multimodal Large Language Models (MLLMs) has recently empowered users to create GUI Agents capable of automating interactions with user interfaces to meet specific user demands  Lu et al. (2024b); Qin et al. (2025); Hong et al. (2023). Nevertheless, determining the optimal strategy for facilitating accurate interaction between agents and GUIs remains a significant challenge.

Early approaches utilizing Set-of-Mark (SoM) identified candidate buttons and click locations on graphical interfaces Yang et al. (2023). Despite their functionality, these methods limited decision space and were prone to missed or false detections, causing interaction inaccuracies. Besides, some solutions attempted to interact directly through raw source code (e.g., HTML, APIs) Furuta et al. (2024); Lù et al. (2024), but these approaches lack intuitive visual grounding, hindering natural graphical interface interaction. Most recently, the interaction mode has shifted focus to vision-based strategies, allowing agents to use visual inputs and text outputs for GUI operations Hong et al. (2023). This approach bypasses earlier constraints by letting agents analyze interface regions freely and align with visual elements intuitively.

Despite these improvements, precise interaction through agent reasoning alone remains a challenge. To address this, our work introduces a location-aware preference optimization approach designed to enhance high-precision GUI interactions.

**GUI Agent Grounding**   The accurate grounding ability of GUI agents, based on visual perception, is crucial for precise interaction. Recently, methods such as those by Gou et al. Gou et al. (2025) and Cheng et al. Cheng et al. (2024) have attempted to learn GUI grounding capabilities directly through Supervised Fine-Tuning (SFT). However, this process often involves challenges related to data format alignment and unclear physical information, making it difficult to achieve more precise localization performance.

In this paper, to enhance the interactive capabilities of GUIs, we explore the use of reinforcement learning to focus the model on exploring the GUI grounding space without interference from other learning processes. We propose a reward mechanism to describe the physical positioning of GUI grounding.

**Preference Optimization in GUI Agents**   Recently, various preference optimization strategies have emerged as significant tools in GUI Agents. Qin et al. Qin et al. (2025) introduced Direct Preference Optimization (DPO) using positive and negative samples from interaction paths to amend erroneous interactions. However, this requires manual construction of sample pairs, which can be labor-intensive and limiting. Xia et al. Xia & Luo (2025) and Lu et al. Lu et al. (2025) developed Rule-based Preference Optimization to assess the accuracy of predicted interaction actions. In contrast, Zhang et al. Zhang et al. (2023) and Liu et al. Liu et al. (2025) employed bounding box positions with fixed threshold constraints to differentiate positive and negative examples. Despite their effectiveness in evaluating interaction accuracy, these approaches commonly rely on static decision boundaries, which offer only coarse evaluations of spatial relationships, leading to imprecise interaction localization.

To address these limitations, we propose Location Preference Optimization (**LPO**), which employs dynamic distance rewards. By directly utilizing positional distance, this approach allows for more precise assessments of interaction relationships across varying locations, enhancing the precision of GUI engagements.

## 3   PROBLEM FORMULATION

The interaction of a GUI Agent can be effectively modeled using the Markov Decision Process (MDP), where the agent perceives and reacts to user inputs to make sequential decisions, as shown in Eq. 1,

$$\mathbf{P}(\langle s_t, a_t \rangle \mid \{\langle s_i, a_i \rangle\}_{i=1}^{t-1}, \mathcal{I}), \tag{1}$$

where $\mathbf{P}(\cdot)$ represents the likelihood of reaching the state-action pair ($\langle s_n, a_n \rangle$) given the preceding sequence ($\{\langle s_i, a_i \rangle\}_{i=1}^{t-1}$) and instruction ($\mathcal{I}$).

The state, $s_t \in \mathbb{R}^{C \times H \times W}$ is represented as an RGB image, capturing the current interface's visual content. The action $a_t$ consists of the tuple ($\mathcal{A}_t \times \mathcal{E}_t$), detailing the agent's strategy. Here, $\mathcal{A}_t$ refers to the interaction action type, such as `click`, `drag`, and `scroll`; $\mathcal{E}_t$ specifies the operation coordinates, which can be a group of points $\{(x^k, y^k)\}_{k=0}^K$, such as bounding box $(x^0, y^0, x^1, y^1)$ or single point $(x^0, y^0)$.

**Optimization Goal**   To enable precise control, our expectation is to maximize the rewards obtained by the GUI agent in the environment at each transition. Therefore, our optimization objective is formulated as Eq. 2,

$$\max_{\theta} \mathbb{E}_{\pi_\theta(a_t|s_t)}[\mathbf{R}(\langle s_t, a_t \rangle)], \tag{2}$$

where $\pi_\theta(a_t|s_t)$ is the probability of selecting action $a$ given state $s$, and $\mathbf{R}(\cdot)$ is the reward obtained from action $a_t$ in state $s_t$.

However, constructing a reasonable reward function remains an important challenge, especially when it is critical that the operation coordinates $\mathcal{E}$ are close in distance. This proximity ensures precise spatial interactions within the GUI, which is essential for achieving optimal performance.

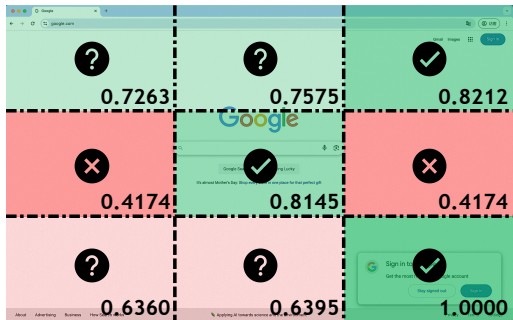 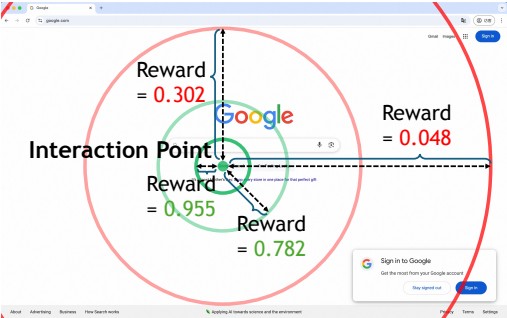

Figure 2: Example of $r_w$. Green zones indicate high interaction likelihood due to rich information, earning greater rewards. In contrast, red zones, like blank areas, have lower interaction probability and rewards. Key interactive areas, such as login, search, and editing zones, align with user interaction tendencies.

Figure 3: Example of $r_d$. When users need to interact at a point located on the search button, the reward increases as the generated interaction point gets closer to this target point, while it decreases as the point moves further away. This highlights the importance of precision in interaction positioning.

## 4 METHODOLOGY

To achieve more precise GUI interactions, although previous approaches Lu et al. (2025); Xia & Luo (2025); Liu et al. (2025) have utilized physical rewards based on interaction space (e.g., IoU or fixed decision boundary), the assessment of rewards for positions remains imprecise (as discussed in Section 2).

**Overview** In this paper, we propose **L**ocation **P**reference **O**ptimization (**LPO**), a novel approach that precisely leverages accurate locational data for preference optimization. **Firstly**, considering that users are more inclined to interact in zones with higher information densities, we segment the interface into distinct windows and utilize their information entropy for a preliminary forecast of interaction positions (Section 4.1). **Secondly**, to provide a finer representation of varying importance across interaction positions, we utilize physical distance to construct a location-based reward metric (Section 4.2). **Lastly**, by amalgamating these rewards, we introduce **LPO**, grounded in the Group Relative Preference Optimization (GRPO) Shao et al. (2024). This approach facilitates a more broader exploration of expansive GUI spaces and directs the agent to optimize towards preferences aligned with precise interaction capabilities (Section 4.3).

### 4.1 WINDOW-BASED INFORMATION DENSITY REWARD

In GUI interaction tasks, an agent iteratively observes the current visual state $s_t \in \mathbb{R}^{C \times H \times W}$, executes an action $a_t \in (\mathcal{A}_t \times \mathcal{E}_t)$, and transitions to the subsequent state $s_{t+1}$ following the trajectory $s_t \rightarrow a_t \rightarrow s_{t+1}$. The distribution of visual information across the interface is heterogeneous. Functional elements like buttons and text fields cluster in regions of high information density. To steer the agent's focus towards these critical regions, we introduce a window-based information density reward.

**Adaptive Window Partition** Firstly, we divide $s_t$ into $K = M \times N$ non-overlapping rectangular windows using a grid resolution of $M$ rows and $N$ columns, as Eq. 3,

$$\mathbf{W}_{i,j} = s_t \left[ :, \frac{(i-1)H}{M} : \frac{iH}{M}, \frac{(j-1)W}{N} : \frac{jW}{N} \right], \quad \forall i \in \{1, \dots, M\}, j \in \{1, \dots, N\}, \quad (3)$$

where $\mathbf{W}_{i,j}$ denotes the window at grid position $(i, j)$. To ensure consistent visual perceptual capacity across the windows in one image, we empirically maintain $M$ and $N$ to match the same settings used by multi-modal large language models.

**Window-wise Entropy Computation** For each window $\mathbf{W}_{i,j}$, we compute its information entropy $\mathcal{H}_{i,j}$ based on the distribution of pixel intensities. Let $p_b(\mathbf{W}_{i,j})$ denote the normalized histogram

probability for pixel intensities within bin $b$. The entropy is calculated as Eq. 4,

$$\mathcal{H}_{i,j} = -\sum_{b=1}^{B} p_b(\mathbf{W}_{i,j}) \log_2 p_b(\mathbf{W}_{i,j}), \tag{4}$$

where $B$ is the total number of bins in the histogram. This entropy measure quantifies the amount of information or uncertainty present in the window's pixel intensity distribution across all grid positions $(i, j)$.

**Reward Formulation** Finally, we map interaction coordinates $(x, y)$ from action $a_t$ to their containing window $\mathbf{W}_{i^*,j^*}$ and normalized entropy values to assign rewards, as Eq. 5,

$$r_w = \frac{\mathcal{H}_{i^*,j^*}}{\max_{i,j} \mathcal{H}_{i,j} + \epsilon}, \quad \text{where} \begin{cases} i^* = \lceil \frac{y}{H/M} \rceil \\ j^* = \lceil \frac{x}{W/N} \rceil \end{cases}, \tag{5}$$

with $\epsilon = 1e - 6$ ensuring numerical stability for low-entropy states.

This reward function directs agents to engage with information-rich GUI elements, like buttons and texts, enhancing interaction accuracy by focusing on zones with higher entropy.

### 4.2 DYNAMIC LOCATION REWARD

While the window-based reward encourages exploration of information-rich regions, precise task execution also requires accurate targeting of specific coordinates. To this end, we introduce a dynamic location reward that directly measures spatial accuracy.

To improve both the accuracy of action types and the precision of operation coordinates $(\mathcal{A}_t \times \mathcal{E}_t)$ in GUI interactions, where $\mathcal{E}_t$ defines operation coordinates as a set of points $\{(x^k, y^k)\}_{k=0}^{K}$, we implement a reward based on physical location. This approach directly incentivizes the agent to perform actions that are spatially accurate, aiming for effective interaction execution.

**Per-Point Reward Formulation** Initially, we calculate the Euclidean distance between each executed coordinate $(x^{*k}, y^{*k})$ in the agent's action set and the corresponding target coordinates $(x^k, y^k)$ in this step. For each pair, we derive a precision reward, as Eq. 6,

$$r_k = \max\left(0, \ 1 - \frac{\sqrt{(x^k - x^{*k})^2 + (y^k - y^{*k})^2}}{d_{max}}\right), \quad \forall k \in \{1, \ldots, K\}, \tag{6}$$

where $d_{max}$ represents the maximum allowable distance used for scaling the reward, set at $1000$.

**Action-Type Constrained Averaging** Subsequently, rewards from individual points are aggregated only when the action type executed by the agent matches the ground truth, as Eq. 7,

$$r_d = \begin{cases} \frac{1}{K} \sum_{k=1}^{K} r_k, & \text{if } \mathcal{A}_t = \mathcal{A}^* \\ 0, & \text{otherwise} \end{cases}, \tag{7}$$

where $\mathcal{A}^*$ is each output action type.

With this reward, agents are strongly encouraged to align their actions with both spatial accuracy across multiple coordinates and the correct action type, thereby fostering efficient and precise GUI interactions.

### 4.3 LOCATION PREFERENCE OPTIMIZATION

To explore a broader space in GUI, based on GRPO Shao et al. (2024), we leverage our location-based reward functions to measure relative location advantages. Our advantage definition is formulated as in Eq. 8,

$$A^{(g)} = \frac{r^{(g)} - \text{mean}(\sum_{g=1}^{G} r^{(g)})}{\text{std}(\sum_{g=1}^{G} r^{(g)})}, \quad r^{(g)} = r_w^{(g)} \times r_d^{(g)}, \tag{8}$$

where $r_w^{(g)}$ and $r_d^{(g)}$ represent the rewards in $g$-times exploitation, and $G$ is the group size. $A^{(g)}$ is the advantage that emphasizes relative position comparison.

After we obtain $A^{(g)}$, we propose the Location Preference Optimization (**LPO**). The policy is updated by maximizing the following objective function as shown in Eq. 9,

$$\mathcal{J}_{\text{LPO}}(\theta) = \mathbb{E}_{\{a_g\}_{g=1}^G \sim \pi_{\theta_{\text{old}}}}$$

$$\frac{1}{G}\sum_{v=1}^{G}\left[\min\left(\underbrace{\frac{\pi_\theta(a_t|s_t)}{\pi_{\theta_{\text{old}}}(a_t|s_t)}}_{\text{Importance Ratio}}A^{(g)}, \text{clip}\left(\frac{\pi_\theta(a_t|s_t)}{\pi_{\theta_{\text{old}}}(a_t|s_t)}, 1-\epsilon_1, 1+\epsilon_2\right)A^{(g)}\right) - \beta\underbrace{\mathbb{D}_{\text{KL}}\left(\pi_\theta \parallel \pi_{\text{ref}}\right)}_{\text{KL Regularization}}\right], \quad (9)$$

$$\mathbb{D}_{KL}\left(\pi_\theta || \pi_{ref}\right) = \frac{\pi_{ref}(a_t|s_t)}{\pi_\theta(a_t|s_t)} - \log\frac{\pi_{ref}(a_t|s_t)}{\pi_\theta(a_t|s_t)} - 1, \quad (10)$$

where $\epsilon_1$, $\epsilon_2$, and $\beta$ are hyperparameters, and $\pi_\theta$ is the policy model to be optimized. For each state $s_t$, we sample a group of actions $\{a_g\}_{g=1}^G$ from the old policy $\pi_{\theta_{\text{old}}}$. The Kullback–Leibler divergence regulation $\mathbb{D}_{\text{KL}}(\cdot)$ controls deviation from the reference model $\pi_{\text{ref}}$.

With this optimization, the GUI Agent's interaction strategy evolves towards more accurate spatial positioning, thereby enhancing its interaction capabilities.

## 5 EXPERIMENTS

This section details the current experimental setup, including the training framework, data, and the baselines used for testing (Section 5.1). Subsequently, we conduct a comprehensive evaluation of our proposed preference optimization method using both offline and online benchmarks (Section 5.2 and Section 5.3). Finally, we validate the effectiveness of our proposed reward function through ablation studies (Section 5.4).

### 5.1 EXPERIMENTAL SETUP

**Training**  Our agent is built upon the foundation model, Ovis2 8B Lu et al. (2024a). During the SFT phase, we employ multiple inner datasets to equip the base model with GUI interaction capabilities. In the RL phase, we employ preference datasets from MMind2Web Deng et al. (2023a), AITZ Zhang et al. (2024b), Omniact Kapoor et al. (2024), OS-Genesis Sun et al. (2024), Mug Li et al. (2022), and GUICourse Chen et al. (2024) to optimize towards more accurate GUI interaction.

**Baselines**  To ensure a fair evaluation, we compare various preference optimization strategies using a single foundation model. Specifically, we select reward functions from UI-R1 Lu et al. (2025) ($R_{UI\_R1}$), GUI-R1 Xia & Luo (2025) ($R_{GUI\_R1}$), and InfiGUI-R1 Liu et al. (2025) ($R_{InfiGUI\_R1}$) as our baselines, each employing distinct preference optimization strategies, as illustrated in Figure 1.

**Computational Resources**  During the preference optimization, the training process lasted approximately 300 GPU hours, under the standard of the NVIDIA H100 GPU [1].

**Hyperparameter Settings**  Following empirical insights from GRPO Shao et al. (2024) and DAPO Yu et al. (2025), we set the learning rate to $1 \times 10^{-6}$ with a constant learning rate scheduler. Additionally, the lower clip range ($\epsilon_1$) is 0.2, while the upper clip range ($\epsilon_2$) is 0.28. The KL regularization hyperparameter ($\beta$) is adjusted to $1 \times 10^{-4}$.

### 5.2 OFFLINE EVALUATION

**GUI Interaction**  We utilized the Multimodal Mind2Web Deng et al. (2023b) benchmark to assess the agent's GUI interaction capabilities. This benchmark is specifically designed to create and evaluate agents' capability to execute arbitrary tasks across various web environments.

---

[1] https://www.nvidia.com/en-sg/data-center/h100/

Table 1: Performance of GUI interaction on Multimodal Mind2Web Deng et al. (2023b). We report Element Accuracy (Ele.Acc), Operation F1 (Op.F1) and Step Success Rate (Step SR). The best model is **in-bold**, and the second best is underlined.

| Method | Cross-Task (↑) | | | Cross-Website (↑) | | | Cross-Domain (↑) | | |
|---|---|---|---|---|---|---|---|---|---|
| | Ele.Acc | Op.F1 | Step SR | Ele.Acc | Op.F1 | Step SR | Ele.Acc | Op.F1 | Step SR |
| **After Supervised Fine-Tuning** | | | | | | | | | |
| Base Modal | 60.3 | 57.4 | 38.2 | 60.7 | 56.9 | 38.4 | 63.8 | 58.5 | 40.7 |
| **After Preference Optimization** | | | | | | | | | |
| + $R_{UI\_R1}$ Lu et al. (2025) | 59.5 | 34.5 | 24.9 | 56.5 | 31.5 | 22.1 | 61.6 | 37.2 | 27.1 |
| + $R_{GUI\_R1}$ Xia & Luo (2025) | 62.5 | 71.6 | 46.6 | 61.2 | 67.6 | 43.5 | 65.0 | 71.1 | 47.9 |
| + $R_{InfiGUI\_R1}$ Liu et al. (2025) | 62.6 | 51.3 | 35.8 | 62.2 | 49.5 | 34.4 | 65.1 | 53.1 | 40.0 |
| + **LPO** (Ours) | **64.3** | **76.7** | **49.5** | **64.4** | **74.4** | **46.4** | **65.2** | **74.8** | **49.6** |

Table 2: Performance of GUI grounding on VisualWebBench Liu et al. (2024). ROUGE-L is used to measure the quality of the generated responses. WebQA is reported by style F1. For other multiple-choice tasks, we report accuracy. The best model is **in-bold**, and the second best is underlined.

| Method | Website (↑) | | | Element (↑) | | Action (↑) | | Average |
|---|---|---|---|---|---|---|---|---|
| | Caption | WebQA | HeadOCR | OCR | Ground | Prediction | Ground | |
| **After Supervised Fine-Tuning** | | | | | | | | |
| Base Model | 23.8 | 77.9 | 69.3 | 96.4 | 96.3 | 96.0 | 91.2 | 78.7 |
| **After Preference Optimization** | | | | | | | | |
| + $R_{UI\_R1}$ Lu et al. (2025) | 23.2 | 78.0 | 69.2 | 96.5 | 96.8 | 96.0 | 91.2 | 78.7 |
| + $R_{GUI\_R1}$ Xia & Luo (2025) | 24.2 | **78.8** | 69.3 | 96.6 | 96.8 | 96.4 | 91.6 | 78.8 |
| + $R_{InfiGUI\_R1}$ Liu et al. (2025) | 23.7 | 75.9 | 69.2 | 96.3 | 96.8 | 96.7 | 91.7 | 78.5 |
| + **LPO** (Ours) | **25.3** | 78.4 | **70.3** | **96.8** | **97.0** | **97.1** | **91.7** | **79.5** |

As shown in Table 1, our preference optimization strategy, **LPO**, significantly outperforms existing models by optimizing GUI interactions through a comprehensive preference optimization approach. **LPO** achieves the highest scores in most metrics across Cross-Task, Cross-Website, and Cross-Domain evaluations. This holistic enhancement underscores **LPO**'s ability to effectively align locational preferences, resulting in more precise and efficient GUI task execution.

**GUI Grounding**  To further evaluate the precise interaction capabilities of agents, we conduct evaluations to determine the effectiveness of preference optimization strategies on enhancing GUI grounding abilities. We employed VisualWebBench Liu et al. (2024) and Screenspot V2 Wu et al. (2024) as benchmarks, providing a broad spectrum of platforms to assess the capacity of GUI agents to accurately ground interaction locations.

VisualWebBench Liu et al. (2024) offers a comprehensive evaluation framework by providing grounding-related tasks in website, element, and action. As shown in Table 2, our experimental results on this benchmark demonstrate that **LPO** consistently achieves SOTA performance and robustness across diverse environments. While GUI-R1 Xia & Luo (2025) shows enhanced WebQA performance, its effectiveness is restricted to particular scenarios and does not improve GUI grounding capabilities across multiple tasks substantially. In contrast, **LPO** shows clear superiority across various metrics, underscoring its robustness and SOTA performance in GUI grounding.

ScreenSpot V2 Wu et al. (2024) provides a benchmark to directly locate text or icons/widgets across different device scenarios, including mobile, desktop, and web environments. As shown in Table 3, our experimental results indicate that **LPO** significantly and comprehensively enhances the visual localization capabilities of the base model across various terminal environments. While GUI-R1 Xia & Luo (2025) and InfiGUI-R1 Liu et al. (2025) outperform **LPO** in a few specific tasks, their overall cross-scenario compatibility is considerably lower, resulting in overall performance that is only comparable to or slightly worse than the base model. In contrast, **LPO** improves upon the base model's performance and achieves SOTA overall results compared to other baselines.

Table 3: Performance of GUI grounding on ScreenSpot V2 Wu et al. (2024). We report grounding accuracy in this table, determining correctness by whether a prediction falls within the ground truth bounding box. The best model is **in-bold**, and the second best is underlined.

| Method | Mobile (↑) | | Desktop (↑) | | Web (↑) | | Average |
|---|---|---|---|---|---|---|---|
| | Text | Icon/Widget | Text | Icon/Widget | Text | Icon/Widget | |
| **After Supervised Fine-Tuning** | | | | | | | |
| Base Model | 97.9 | 80.0 | 94.8 | **86.4** | 93.5 | 84.2 | 89.5 |
| **After Preference Optimization** | | | | | | | |
| + $R_{UI\_R1}$ Lu et al. (2025) | 97.5 | 77.7 | 93.8 | 82.1 | 94.0 | 84.2 | 88.2 |
| + $R_{GUI\_R1}$ Xia & Luo (2025) | 97.5 | 77.7 | 94.8 | 84.2 | 93.5 | **84.7** | 88.7 |
| + $R_{InfiGUI\_R1}$ Liu et al. (2025) | **98.2** | 80.0 | 95.3 | 86.0 | 93.5 | 83.2 | 89.5 |
| + **LPO** (Ours) | 97.9 | **82.9** | 95.9 | 86.4 | 95.6 | 84.2 | **90.5** |

Table 4: Performance of online evaluation on WebVoyager He et al. (2024). We report the Task Success Rate in the table. The best model is **in-bold**, and the second best is underlined.

| | Amazon | Apple | ArXiv | BBC News | Coursera |
|---|---|---|---|---|---|
| **After Supervised Fine-Tuning** | | | | | |
| Base Model | 40.0 | 58.1 | 53.4 | 38.0 | 54.7 |
| **After Preference Optimization** | | | | | |
| + $R_{UI\_R1}$ Lu et al. (2025) | 12.2 | 41.8 | 51.1 | 30.9 | 45.2 |
| + $R_{GUI\_R1}$ Xia & Luo (2025) | 35.0 | 37.2 | 27.9 | 33.3 | 57.1 |
| + $R_{InfiGUI\_R1}$ Liu et al. (2025) | **51.2** | 51.1 | 55.8 | **59.5** | 69.0 |
| + **LPO** (Ours) | **51.2** | **60.5** | **64.3** | 54.7 | **71.4** |

| | Github | Huggingface | Wolfram Alpha | ESPN | Overall |
|---|---|---|---|---|---|
| **After Supervised Fine-Tuning** | | | | | |
| Base Model | **65.8** | 33.3 | 56.5 | **41.8** | 48.0 |
| **After Preference Optimization** | | | | | |
| + $R_{UI\_R1}$ Lu et al. (2025) | 58.3 | **51.1** | 63.0 | 27.9 | 47.3 |
| + $R_{GUI\_R1}$ Xia & Luo (2025) | 50.0 | 35.0 | 56.5 | 15.9 | 37.5 |
| + $R_{InfiGUI\_R1}$ Liu et al. (2025) | 53.6 | 43.6 | **65.9** | 41.8 | 54.1 |
| + **LPO** (Ours) | 56.1 | 47.5 | 57.5 | 38.6 | **57.6** |

## 5.3 ONLINE EVALUATION

To thoroughly assess the applicability of our preference optimization strategy in real-world scenarios, we conducted online evaluations to directly measure the performance of the GUI Agent in dynamic online environments.

We utilized WebVoyager He et al. (2024) as our benchmark, performing online evaluations on nine accessible websites: Amazon, Apple, Arxiv, BBC News, Coursera, GitHub, Hugging Face, Wolfram Alpha, and ESPN. Other websites were unavailable due to network issues (Google Search and Google Map), timeliness (Booking, Google Flights), and anti-scraping measures (Allrecipes, Cambridge Dictionary).

As shown in the Table 4, our preference optimization strategy enhances the interaction accuracy of GUI Agents in online environment. Although accuracy slight decreasing on a few websites, our strategy achieved SOTA accuracy overall. In contrast, other baselines lack precision measure in position and, despite improvements on certain websites, fail to achieve high performance overall.

Table 5: Performance of ablation study on Multimodal Mind2Web Deng et al. (2023b). The best model is **in-bold**, and the second best is underlined.

| Method | Cross-Task (↑) | | | Cross-Website (↑) | | | Cross-Domain (↑) | | |
|---|---|---|---|---|---|---|---|---|---|
| | Ele.Acc | Op.F1 | Step SR | Ele.Acc | Op.F1 | Step SR | Ele.Acc | Op.F1 | Step SR |
| **After Supervised Fine-Tuning** | | | | | | | | | |
| Base Modal | 60.3 | 57.4 | 38.2 | 60.7 | 56.9 | 38.4 | 63.8 | 58.5 | 40.7 |
| **After Preference Optimization** | | | | | | | | | |
| w/o $r_d$ | 56.7 | 74.6 | 42.3 | 56.3 | 69.7 | 40.9 | 61.2 | 73.1 | 45.6 |
| w/o $r_w$ | 62.7 | 71.7 | 46.4 | 61.6 | 70.3 | 44.1 | 64.2 | 71.9 | 47.6 |
| **+LPO** (Ours) | **64.3** | **76.7** | **49.5** | **64.4** | **74.4** | **46.4** | **65.2** | **74.8** | **49.6** |

## 5.4 ABLATION STUDY

We conduct ablation experiments on our two rewards proposed in this paper and analyze their impact on the overall performance in Table 5 w/o $r_d$ and w/o $r_w$.

**Effectiveness of Window-based Information Density Reward**   To demonstrate the efficacy of the window-based information density reward $r_w$, we compare the performance of our optimization strategy with and without $r_w$. As shown in Table 5 (w/o $r_w$), the absence of $r_w$ leads to a decline in operational accuracy, underscoring the importance of focusing on high-density informational areas to enhance the agent's decisiveness and effectiveness in GUI agent interaction.

**Effectiveness of Dynamic Location Reward**   To validate the effectiveness of the dynamic location reward $r_d$, we similarly compared our performance with and without $r_d$. As indicated in Table 5 (w/o $r_d$), the exclusion of $r_d$ result in a significant reduction in element accuracy due to the absence of spatial relationship. This highlights the substantial impact of dynamic location reward on GUI spatial optimization. Additionally, the success rate per action and operational correctness also declined, demonstrating the critical role of location information in action decision-making.

## 6 LIMITATIONS

**Dependence on Extensive High-Precision Location Datasets**   While **LPO** offers significant enhancements, its performance is highly dependent on the availability of large datasets with precise grounding annotations. In situations where these datasets are inadequate or poorly constructed, the system is susceptible to performance degradation. This reliance not only necessitates substantial effort in data collection and annotation but also poses challenges for its practical application and widespread adoption.

**Significant Computational Overhead**   Training the **LPO** approach demands considerable computational power due to its complex integration of locational data and dynamic reward mechanisms. This high computational requirement can hinder real-time application scenarios and limit accessibility to users with less advanced computing resources.

## 7 CONCLUSION

In this paper, we delve into the challenge of achieving high-accuracy interactions for autonomous agents in GUI. We propose a novel solution: Location Preference Optimization (**LPO**). This approach is designed to refine interaction accuracy by utilizing locational data to inform and optimize interaction preferences, thus addressing the shortcomings of existing methodologies. **LPO** significantly improves GUI agents' interaction capabilities, demonstrating superior performance in both offline benchmarks and online evaluations. This advancement sets a new standard for precision in GUI interactions and lays the groundwork for more intelligent and adaptive systems, offering a promising direction for future developments in complex interface interactions.

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

This appendix introduces the social impact and future work of this paper.

## A  SOCIAL IMPACT

The development and deployment of autonomous agents capable of interacting effectively with Graphical User Interfaces (GUIs) have notable social implications. Primarily, these agents significantly reduce labor and time costs associated with manual GUI operations by utilizing natural language processing as an intermediary. This reduction not only enhances productivity in digital environments but also enables a more inclusive digital transformation by allowing individuals with less technical expertise to engage efficiently with complex software systems.

Moreover, the introduction of Location Preference Optimization (**LPO**) addresses essential challenges in spatial localization, potentially leading to more adaptive and intelligent systems. By improving interaction accuracy across diverse environments, **LPO** paves the way for more intuitive user experiences, which could democratize access to advanced technologies and improve equity in digital interactions.

However, the widespread integration of such autonomous systems also raises important ethical considerations. As GUI agents become more prevalent, there's a need to ensure they are used responsibly and do not inadvertently eliminate jobs, particularly those reliant on manual operations. Additionally, safeguarding user data and maintaining privacy during interactions are paramount to preserving trust in these technologies.

Overall, the advancements presented in this research offer significant potential benefits but must be balanced with careful consideration of their broader social and ethical impacts.

## B  FUTURE WORK

While **LPO** has shown significant advancements in GUI interaction capabilities, several avenues for future research could further elevate its potential:

**Enhanced Dataset Diversity**   Expanding the diversity of high-precision datasets used for training and evaluation could improve the robustness of **LPO**. This includes incorporating a variety of GUI designs and interaction patterns from different cultural and professional contexts to ensure wider applicability.

**Real-Time Optimization**   Future efforts could focus on optimizing the computational efficiency of **LPO**, enabling its deployment in real-time applications. Techniques such as model compression or adaptive learning algorithms might be explored to reduce the computational overhead.

**Ethical and Responsible Use**   Further research should also address ethical considerations, focusing on creating guidelines and frameworks to ensure that **LPO** and similar technologies are used responsibly and do not reinforce biases or invade user privacy.

## USAGE OF LARGE LANGUAGE MODELS

In the preparation of this work, the authors used a Large Language Model (LLM) primarily as a tool for language polishing and refinement of the manuscript. The LLM was solely employed to assist in improving the fluency and clarity of the writing. Importantly, the LLM was not utilized in the conceptualization of the research, the development of the methodology, the analysis of results, or the drawing of scientific conclusions. All intellectual contributions and substantive content remain the responsibility of the authors.

