# OpenReview forum: "LPO: Towards Accurate GUI Agent Interaction via Location Preference Optimization"
_ICLR.cc/2026/Conference — ICLR 2026 Conference Withdrawn Submission_

### Official Review · Reviewer_NSyy · 2025-10-23

**Soundness:** 2
**Presentation:** 3
**Contribution:** 2
**Rating:** 4
**Confidence:** 4

**Summary:**

This paper introduces Location Preference Optimization (LPO), a reinforcement learning framework for improving GUI agent interactions by optimizing spatial accuracy. The method defines two additional reward functions — (1) a window-based information density reward based on image entropy, and (2) a dynamic location reward based on the Euclidean distance between predicted and ground-truth coordinates. These rewards are integrated into a GRPO (Group Relative Preference Optimization) structure to fine-tune multimodal LLM-based GUI agents. The authors claim that LPO achieves state-of-the-art results on several GUI interaction and grounding benchmarks such as Mind2Web, VisualWebBench, ScreenSpot V2, and WebVoyager.

**Strengths:**

1. The paper is well-written and clearly organized. Each component of LPO is systematically introduced with visual illustrations

2. The experimental section is extensive, including multiple offline and online benchmarks as well as ablations.

3. Reported results show consistent gains across several metrics and datasets, indicating that the method is at least empirically effective.

**Weaknesses:**

1. The proposed LPO framework builds upon GRPO and mainly introduces two additional handcrafted reward functions. While these rewards are intuitively reasonable and practically useful, they appear to be straightforward extensions of existing concepts in spatial reasoning and reinforcement learning. The paper would benefit from clarifying the conceptual novelty or theoretical insight beyond this combination.

2. The claim that SFT or prior RL methods "cannot assess positional accuracy effectively" is not strongly substantiated. The paper fails to provide an analysis or diagnostic experiment showing why existing reward formulations fail.

**Questions:**

no question

---

### Official Review · Reviewer_2MDH · 2025-10-28

**Soundness:** 3
**Presentation:** 3
**Contribution:** 3
**Rating:** 6
**Confidence:** 3

**Summary:**

This paper presents LPO, a new approach for improving how autonomous agents interact with GUIs using natural language. Existing methods based on SFT and RL struggle with precise positional understanding, which affects interaction accuracy. LPO addresses these limitations by utilizing information entropy to focus on informative regions of the interface and employing a dynamic reward based on physical distance to better evaluate interaction positions. Supported by GRPO, LPO enables more comprehensive exploration and optimization within GUI environments. Experimental results show that LPO achieves superior performance on offline benchmarks and online evaluations, surpassing previous approaches in GUI interaction and grounding tasks.

**Strengths:**

- The paper proposes an comprehensive and innovative approach to improving spatial accuracy for GUI agents. By integrating window-based information entropy and a dynamic location-based reward, the methodology provides a precise and context-sensitive scheme for guiding agent interactions. This dual-reward design tackles key shortcomings in existing spatial localization strategies, which often relied on coarser IoU thresholds or rigid boundary rules. Although the idea is simple itself, it works for GUI agents's spatial accuracy.
- The use of GRPO as an optimization backbone facilitates extensive exploration and fine-grained preference tuning, enhancing both the breadth and depth of agent learning. Another major strength is the rigorous evaluation across both offline and online benchmarks, where the proposed framework consistently achieves strong, state-of-the-art performance.
- The paper’s mathematical clarity and well-structured methodology make its contributions easier to understand. The practical alignment with natural language-driven interfaces emphasizes its relevance for real-world multimodal agents.

**Weaknesses:**

- I strongly suggest that the authors address the citation problems present in the original paper. Proper and consistent citation is essential for both the credibility of the work and for guiding readers to relevant prior research. When the author's name is part of your sentence, include only the year in parentheses after the name. When the author and year are both in parentheses, put both together in "()".
-  Some design decisions, such as the choice of grid resolution for window partitioning and the parameters used for entropy computation, are set empirically and may require further adaptation for different GUI environments or tasks. This reliance on heuristics can limit the method’s out-of-the-box generalizability. How does the choice of grid resolution (M, N) for window partitioning affect the performance of the information entropy-based reward?
- The dependence on exact action-type matching in the reward calculation may also magnify penalties for misclassification, which could undermine improvements in spatial precision if the agent struggles to identify the correct action type.

**Questions:**

- What strategies are used to compute the normalized histogram for entropy calculation, and how sensitive is the entropy measure to the number of bins (B)?
- When aggregating per-point rewards, how are multiple targets with overlapping spatial coordinates handled to avoid over-penalization?
- Does the methodology support non-rectangular or dynamically resizable GUI window partitions?

---

### Official Review · Reviewer_J5vV · 2025-10-30

**Soundness:** 2
**Presentation:** 3
**Contribution:** 2
**Rating:** 4
**Confidence:** 2

**Summary:**

This paper tackles precise spatial localization for GUI agents by introducing LPO. LPO uses a novel reward function combining information entropy to find important zones and a dynamic location reward based on physical distance for precision. This reward, optimized via GRPO, achieves SOTA results on multiple GUI benchmarks.

**Strengths:**

- Intuitive design. Using physical distance as a continuous precision signal is a clear improvement over prior coarse, binary rewards.
- Strong empirical performance, achieving SOTA results across a comprehensive suite of offline and online benchmarks .
- Clear writing.

**Weaknesses:**

- My main concern is the core assumption for the $r_w$ reward: that high information entropy correlates with interaction likelihood 1. This heuristic seems brittle and non-generalizable. In many realistic scenarios, the opposite can be true;  For example, in a minimalist UI, a critical "Submit" button might be a simple, large, solid-color block (very low entropy). Conversely, a non-interactive, complex advertisement banner on the same page would have very high entropy. In this common case, $r_w$ would actively penalize the correct action and reward focusing on the distractor. This suggests the $r_w$ component may be learning a spurious correlation specific to the benchmark data.

- The paper mentions "Significant Computational Overhead". If LPO is an order of magnitude more expensive than baselines for a modest accuracy gain, I believe the overall contribution is limited. Specifically, how much more computationally expensive is it (for training and inference, respectively)? And for each, are there potential ways to reduce this cost?

**Questions:**

- (from weakness 1) Have the authors tested robustness in scenarios where entropy negatively correlates with the target (e.g., minimalist UIs vs. complex, non-interactive backgrounds)?

- Why was a pixel-level entropy metric chosen over a more semantic, learned approach? For example, the vision model's own semantic features could be used to predict a general "interactability map" (e.g., a map predicting the likelihood of a region being a button, link, or text field). While this would require training on an annotated dataset (potentially necessitating new labeling effort if existing datasets are insufficient), this learned $r_w$ would seem to solve the heuristic's core weakness and be far more generalizable.

---

### Official Review · Reviewer_fcd7 · 2025-11-01

**Soundness:** 3
**Presentation:** 3
**Contribution:** 2
**Rating:** 4
**Confidence:** 4

**Summary:**

The current paper proposes a method for fine-tuning GUI agents, focusing on interaction positions, using a novel reward function in a GRPO-inspired loss.

Precisely, the reward is the product of an inverse distance to target bonus and a reward derived from the information entropy of the window (patch) that contains that location.

The intuition is that the agent should be rewarded for focusing on areas rich in information.

**Strengths:**

- clear presentation
- strong empirical results

**Weaknesses:**

- given that the novelty comes mainly from the information-

**Questions:**

1. Isn't this method actually using standard GRPO, and in fact just proposing a novel reward function?
2. Is the histogram of colours really a measure of how rich in interactive elements the area is? For example, an image would be a big distractor for this metric. Or 10 small buttons or a large one could lead to the same pixel value histogram.

---

### Official Review · Reviewer_JXud · 2025-11-03

**Soundness:** 3
**Presentation:** 3
**Contribution:** 2
**Rating:** 4
**Confidence:** 2

**Summary:**

This paper introduce Location Preference Optimization (LPO) , an approach to leverages locational data to optimize interaction preferences. LPO employs Group Relative Preference Optimization (GRPO) for preference optimization.

Reward design is designed in two parts, 1) information entropy, to predict interaction positions by focusing on zones rich in information; 2) a dynamic location reward function based on physical distance, reflecting the varying importance of interaction positions. Comprehensive experiments demonstrate LPO’s superior performance, achieving SOTA results across both offline benchmarks and real-world online evaluations.

**Strengths:**

1. The method is simple but effective with strong generalization across different tasks.

2. The reward design raises the information entropy metric, which is effective according to author’s experiments, but ignored by previous methods.

**Weaknesses:**

Key details were somewhat lacking in methodology and experiment parts - please see questions section below.

**Questions:**

1. In section 4.1, the screenshot is divided into M times N non-overlapping rectangles, but the settings of hyper-parameters M, N should be discussed.

2. In fig. 2, why the bottom-right has a 1.0 entropy score. What is “Key interactive areas, such as login, search, and editing zones, align with user interaction tendencies” means and how to achieve that.

3. In section 4.2, Euclidean distance refers to the same reward along with the circle, while most of the icon is designed as rectangle. The contradiction should be discussed.

4. In section 4.2, why there is an averaging operation across K coordinates, since each step only has single ground-truth.

5. In section 5.1, most training details should be discussed, like the type of training data in SFT phase, the data composition and dynamics during RL training.

6. All the experiments are built upon the fine-tuned Ovis-2-8B, is there any special design in SFT phrase before LPO, could other models in different architecture benefit from LPO, like Qwen or Intern family?

---

### Note · Authors · 2025-12-30

I have read and agree with the venue's withdrawal policy on behalf of myself and my co-authors.